# Early Reoperation Rates and Its Risk Factors after Instrumented Spinal Fusion Surgery for Degenerative Spinal Disease: A Nationwide Cohort Study of 65,355 Patients

**DOI:** 10.3390/jcm11123338

**Published:** 2022-06-10

**Authors:** Jihye Kim, Hwan Ryu, Tae-Hwan Kim

**Affiliations:** 1Division of Infection, Department of Pediatrics, Kangdong Sacred Heart Hospital, Hallym University College of Medicine, Seoul 05355, Korea; jihyewiz17@kdh.or.kr; 2Spine Center, Department of Orthopedics, Hallym University Sacred Heart Hospital, Hallym University College of Medicine, Anyang 14068, Korea; ryuhwan@hallym.or.kr

**Keywords:** reoperation, risk factor, prediction, spinal fusion, instrumentation, surgical approach, degenerative spinal disease, big data

## Abstract

Reoperation is a major concern in spinal fusion surgery for degenerative spinal disease. Earlier reported reoperation rates were confined to a specific spinal region without comprehensive analysis, and their prediction models for reoperation were not statistically validated. Our study aimed to present reasonable base rates for reoperation according to all possible risk factors and build a validated prediction model for early reoperation. In our nationwide population-based cohort study, data between 2014 and 2016 were obtained from the Korean National Health Insurance claims database. Patients older than 19 years who underwent instrumented spinal fusion surgery for degenerative spinal diseases were included. The patients were divided into cases (patients who underwent reoperation) and controls (patients who did not undergo reoperation), and risk factors for reoperation were determined by multivariable analysis. The estimates of all statistical models were internally validated using bootstrap samples, and sensitivity analyses were additionally performed to validate the estimates by comparing the two prediction models (models for 1st-year and 3rd-year reoperation). The study included 65,355 patients: 2939 (4.5%) who underwent reoperation within 3 years after the index surgery and 62,146 controls. Reoperation rates were significantly different according to the type of surgical approach and the spinal region. The third-year reoperation rates were 5.3% in the combined lumbar approach, 5.2% in the posterior lumbar approach, 5.0% in the anterior lumbar approach, 3.0% in the posterior thoracic approach, 2.8% in the posterior cervical approach, 2.6% in the anterior cervical approach, and 1.6% in the combined cervical approach. Multivariable analysis identified older age, male sex, hospital type, comorbidities, allogeneic transfusion, longer use of steroids, cages, and types of surgical approaches as risk factors for reoperation. Clinicians can conduct comprehensive risk assessment of early reoperation in patients who will undergo instrumented spinal fusion surgery for degenerative spinal disease using this model.

## 1. Introduction

Generally, revision spine surgery for degenerative spinal disease is more invasive than primary surgery, and its surgical technique is also more demanding. Therefore, revision spinal surgery shows worse clinical outcomes than primary surgery in terms of surgical complications [1,2,3]. If patients undergo invasive primary surgery, the risk of requiring revision surgery sharply increases [4]; more invasive revisional surgery can lead to detrimental outcomes, including mortality [5]. Numerous studies have investigated reoperation rates following spinal surgery for degenerative spinal disease and its risk factors.

The risk of reoperation is influenced by numerous factors, including the patients’ symptoms, radiological findings of the disease, and diverse factors, including lifestyle, health insurance, surgeon preference, and contemporary treatment trends. Some of these factors cannot be easily quantified, and they inevitably act as confounders in studies investigating reoperation risks. It is impossible to avoid the effects of these confounders; hence, researchers generally adopt one of the following two strategies: the first strategy is to report the reoperation rates of study cohorts with a large sample size, and the second is to build a prediction model for reoperation using all possible information. To successfully implement these strategies, studies reporting reoperation rates should be based on the general population, and studies on building a prediction model for reoperation should be statistically validated considering inevitable biases from confounders. However, these two conditions have not been achieved in previous studies. The reported reoperation rates based on limited samples were inevitably confined to a specific spinal region, mainly the cervical or lumbar regions, without a comprehensive analysis including all spinal regions [3,4,6,7,8,9]. Therefore, reoperation rates and risk factors vary considerably according to the characteristics of their study patients and design [7,9,10,11]. Discrepancies in the estimates of unvalidated prediction models exponentially increase when calculating the risk factors for long-term reoperation [7,12,13,14,15].

To overcome the limitations of previous studies, we conducted a population-based cohort study using nationwide claims data. Our study has two distinct research purposes. First, using a database including the entire population and their surgical records across all spinal regions, we aimed to present reasonable ‘base rates’ for reoperation according to all possible risk factors obtained from the database. The risk for morbidity from reoperation sharply increases when primary surgery is more invasive, and reoperation rates in patients who undergo more invasive primary surgery are clinically important. Therefore, we selected patients who had undergone instrumented fusion surgery for degenerative spinal disease. Second, based on these population cohorts, we tried to build a validated prediction model for early reoperation, since a prediction model for long-term reoperation is greatly influenced by numerous unknown confounders and is theoretically incapable of making accurate predictions.

## 2. Materials and Methods

### 2.1. Database

In this nationwide population-based cohort study, data were obtained from the Health Insurance Review and Assessment Service (HIRA) database. The HIRA database contains all inpatient and outpatient data from hospitals and community clinics in Korea, allowing for the inclusion of a nationwide cohort study of the entire population. Diagnostic codes were assigned according to the modified version of the 10th revision of the International Classification of Diseases (ICD-10) and the 7th revision of the Korean Classification of Diseases (KCD-7). Drug use under diagnosis was identified using anatomical therapeutic chemical codes and the HIRA general name codes. This study was approved by the Institutional Review Board of our hospital (IRB No. 2020-03-009-001).

### 2.2. Study Patients

We included patients aged >19 years who underwent instrumented spinal fusion surgery (index surgery) for degenerative spinal disease (index disease) between 1 January 2014 and 31 December 2016. Degenerative spinal diseases were identified using the following codes: spinal stenosis (M48.0), spondylolisthesis (M43.1), spondylolysis (M43.0), other spondylosis (M47.1 and M47.2), and cervical disc disorder (M50). Methods of spinal instrumentation in the index surgeries were identified using the following electronic data interchange codes: anterior cervical approach (N2461, N0464, and N2463), posterior cervical approach (N2467, N2468, N0467, and N2469), anterior thoracic approach (N0465, N2464, N2465, and N2466), posterior thoracic approach (N0468), anterior lumbar approach (N0466 and N1466), and posterior lumbar approach (N0469, N1460, N1469, and N2470).

We excluded patients who had undergone previous spinal surgeries within 2 years before the index disease and those who were treated under the ICD-10 codes of spinal infection (A18.00, M46, M49, G06, and T814), spine fractures (S1, S2, S3, T02.0, T02.1, T02.7, T08, T09, T91, M48.3, M48.4, and M48.5), or malignancy (C, Figure 1). We also excluded patients who underwent spinal fusion surgeries in multiple spinal regions at the time of the index surgery. Patients with incomplete data were also excluded, and a minimum follow-up period of 3 years was mandatory for study inclusion.

### 2.3. Definitions of Reoperation

Reoperation was defined as a surgical treatment that was performed in the same region after more than 30 days from the index surgery (Figure 2) and included the fusion surgeries mentioned above and the following decompressive surgeries: cervical decompressive surgery (N2491, N2492, N0491, N1491, N1497, and N2497), thoracic decompressive surgery (N1492, N1498, and N2498), and lumbar decompressive surgery (N0492, N1493, N1499, and N2499). In our country, patients who are scheduled to undergo ‘staged surgery’, including combined anterior and posterior approaches, often undergo their second-stage operation 1–3 weeks after the first. Therefore, a 30-day period was set up to identify the ‘staged surgery’. In addition, reoperation was defined as subsequent surgery, which only included decompressive or fusion surgery, and we tried to exclude subsequent surgeries to correct minor failures, such as misplaced screws.

### 2.4. Covariates

Data on demographic characteristics and the type of hospital in which the index surgery was performed were retrieved. The hospitals were categorized as tertiary, general, or other hospitals. Tertiary hospitals are general hospitals that are approved to provide most types of advanced medical care, treat severely ill patients, and have a minimum of 20 departments. General hospitals were defined as those with more than 100 beds and more than seven or nine medical specialties. Preexisting medical comorbidities appearing within 1 year before the index surgery were identified according to the ICD-10 codes (Appendix A) and evaluated using the Charlson comorbidity index (CCI). The CCI score is the sum of the weighted scores for each comorbidity and shows good agreement with the ICD-10 codes [16,17]. Data regarding transfusion (allogenous or autologous; Appendix A) and steroid use (Appendix A) during the index surgery were also retrieved.

### 2.5. Statistical Analysis

Data are reported as mean ± standard deviation for numerical variables and number of patients and as frequencies (%) for categorical variables. Annual reoperation rates according to the spinal region were calculated with 95% CIs. We also calculated annual reoperation rates in relation to various patient characteristics and precise surgical procedures.

Logistic regression models were used to identify risk factors for reoperation within 3 years after instrumented fusion surgery for degenerative spinal disease. All significant independent variables (*p* < 0.05) from the univariable analysis were included in the multivariable model. Multicollinearity between covariates was tested using a variance inflation factor. The performance of the prediction model was assessed by the area-under-the-receiver-operating-characteristic curve for discriminative ability and Hosmer–Lemeshow goodness-of-fit statistics for calibration.

The prediction model for reoperation was validated using the following two-step procedure: First, the estimates of all prediction models were internally validated with relative bias based on 1000 bootstrapped samples. Second, a sensitivity analysis was performed to identify the risk factors for reoperation at a different time point: within 1 year after the index surgery. We then assessed potential effect modification by unknown confounders by comparing the adjusted odds ratios (ORs) and 95% Cis of the consistent risk factors that remained as significant predictors in the two prediction models. The adjusted ORs were compared using the Bland and Altman interaction tests. Data extraction and statistical analysis were performed using the SAS Enterprise Guide 6.1 (SAS Institute, Cary, NC, USA).

## 3. Results

We identified 125,113 patients who underwent instrumented spinal fusion surgery (index surgery) for degenerative spinal disease (index disease) between 2014 and 2016 (Figure 1). Among them, we excluded patients who had undergone spinal surgeries within 2 years before the index surgery (*n* = 8154), those who were treated under the ICD-10 codes of spine infection, spine fracture, and malignancy within 2 years before the index surgery (*n* = 50,999), those who underwent spinal fusion surgeries in multiple regions at the time of the index surgery (*n* = 356), and those who had missing data (*n* = 249).

The mean age of the remaining 65,355 patients was 61.4 years (standard deviation, 11.5 years), and 55% (35,749) were women. The median interval between index surgery and reoperation was 520 days (interquartile range, 232–800 days).

### 3.1. Annual Reoperation Rates according to the Spinal Regions

Among the 65,355 patients, reoperation was performed in 1.7% (*n* = 1105) at 1-year follow-up and in 4.5% (*n* = 2939) at 3-year follow-up (Table 1). The proportions of the operated spinal regions were 26% (16,962 patients), 2% (1186 patients), and 72% (47,207 patients) in the cervical, thoracic, and lumbar regions for the spine, respectively (Table 1). The annual reoperation rates and their 95% CIs according to the spinal regions are presented in Table 1 and Figure 3. Reoperation rates at 3-year follow-up were 2.7% (95% CI, [2.5–3.0]), 3.2% (95% CI, [2.2–4.2]), and 5.2% (95% CI, [5.0–5.4]) at the cervical, thoracic, and lumbar regions fo the spine, respectively. Throughout the follow-up period, reoperations were the most common in the lumbar spine.

### 3.2. Annual Reoperation Rates according to Patients’ Characteristics

The annual reoperation rates according to patient characteristics are presented in Table 2. Although 1st-year reoperation rates were the highest (2.2%) in the oldest age group (age > 80 years), 3rd-year reoperation rates were the highest (5.3%) in patients aged between 70 and 79 years. Reoperation rates were higher in male patients or those living in a rural residence, in those who underwent index surgery at general or other-type hospitals, and in those who received blood transfusion or had a longer duration of steroid therapy.

### 3.3. Annual Reoperation Rates according to Comorbidities

Patients with severe medical comorbidities according to the CCI score consistently showed higher reoperation rates (Table 3). Reoperation rates were noticeably higher in patients with renal disease, and patients with end-stage renal disease showed the highest reoperation rates in the 1st (6.6%) and 2nd year (8.5%) after the index surgery. Patients with Parkinson’s disease also showed consistently higher reoperation rates (3.8%, 7.6%, and 9.6% in the 1st, 2nd, and 3rd years, respectively).

On the other hand, although the patients with moderate to severe liver disease showed a lower 1st-year reoperation rate (1.6%) than the overall rate (1.7%), their reoperation rates sharply increased, and they showed the highest 3rd-year reoperation rate (11.5%). In addition, first-year reoperation rates were especially high in patients with congestive heart failure (2.9%) and complicated diabetes (2.8%), and 3rd-year reoperation rates were higher in patients with rheumatological disease (7.5%).

### 3.4. Annual Reoperation Rates according to the Surgical Procedures

Among the 2939 patients who underwent reoperation 3 years after the index surgery, 1882 (64%) underwent revisional fusion surgery as the first reoperation, and the remaining 1059 (36%) underwent revisional decompressive surgery without fusion as the first reoperation (Table 4).

The most common method for index surgeries was the posterior lumbar approach (63%; 41,259 of 65,355; Table 4). The two least common methods were combined thoracic (0.04%, 27 cases) and anterior thoracic (0.05%, 34 cases) approaches. Except for the two least common approaches, 3rd-year reoperation rates were the highest in the combined lumbar approach (5.3%; 95% CI, [4.6–5.9]), followed by the posterior lumbar approach (5.2%; 95% CI, [5.0–5.4]). Also 3rd-year reoperation rates were the lowest in the combined cervical approach (1.6%; 95% CI, [0.3–2.9]).

### 3.5. Risk Factors for Early Reoperation within 3 Years after Instrumented Spinal Fusion Surgeries: Multivariable Analysis

Multivariable analysis using the backward elimination method identified the following variables as predictors for reoperation within 3 years after the index surgery (model 2 in Table 5): age between 50–69 years (OR: 1.32 [1.15–1.51]), age between 70–79 years (OR: 1.30 [1.12–1.51]), male sex (OR: 1.66 [1.53–1.80]), general (OR: 1.33 [1.20–1.48]) or other (OR: 1.12 [1.02–1.23]) hospital types, congestive heart failure (OR: 1.29 [1.08–1.54]), chronic pulmonary disease (OR: 1.20 [1.10–1.30]), rheumatologic disease (OR: 1.69 [1.47–1.94]), peptic ulcer disease (OR: 1.21 [1.10–1.33]), moderate to severe liver disease (OR: 2.41 [1.09–5.34]), uncomplicated (OR: 1.14 [1.05–1.26]) and complicated diabetes (OR: 1.26 [1.10–1.44]), osteoporosis (OR: 1.12 [1.01–1.25]), Parkinson’s disease (OR: 2.10 [1.56–2.82]), end stage renal disease (OR: 2.15 [1.45–3.17]), allogenous transfusion (OR: 1.33 [1.23–1.45]), systemic steroid use over 2 weeks (OR: 2.25 [1.33–3.80]), anterior cervical approach (OR: 0.55 [0.48–0.63]), posterior cervical approach (OR: 0.45 [0.31–0.64]), posterior thoracic approach (OR: 0.51 [0.36–0.72]), combined cervical approach (OR: 0.22 [0.09–0.50]), and cage use (OR: 1.13 [1.02–1.24]). Multicollinearity among covariates was low, and all variance inflation factors were less than 1.9. The Hosmer–Lemeshow goodness-of-fit test indicated good calibration (*p* = 0.266), and the area-under-the-receiver-operating-curve was 0.695.

### 3.6. Validation of the Prediction Model

#### 3.6.1. *Bootstrap Validation*

After bootstrap validation, the relative bias of the estimate was lower, between −2.3% and 9.0%, in age, hospital type, steroid use, and surgical approach type (Table 5). However, the relative bias of the estimate was generally higher, exceeding 31.1% for medical comorbidities, except for uncomplicated diabetes (11.1%) and osteoporosis (5.2%). Bootstrap-adjusted ORs and 95% CIs are shown in Figure 4.

#### 3.6.2. Sensitivity Analysis and Bland-Altman Test for Interaction

Sensitivity analysis was performed to assess our prediction model when early reoperation was defined as operation performed within 1 year after the index surgery (Appendix A). Other types of hospitals, combined cervical approach, and medical comorbidities, including moderate to severe liver disease, chronic pulmonary disease, and osteoporosis, which were significant predictors for the 3rd-year reoperation, were not significant predictors for this prediction model for the 1st-year reoperation. The bootstrap-adjusted ORs and 95% CIs for the prediction model for the 1st-year reoperation are displayed in Figure 5. Potential effect modification by unknown confounders was assessed by comparing the adjusted ORs and 95% CIs of the consistent predictors in the two prediction models (models for 1st-year and 3rd-year reoperation), and are presented in Table 6. No significant effect modification was observed for any variable.

## 4. Discussion

Based on the data from a population cohort of 65,355, 2939 (4.5%; 95% CI, [4.3–4.7]) patients underwent reoperation within 3 years after instrumented spinal fusion surgeries for degenerative spinal disease. Among them, approximately two-thirds (64%, 1882 patients) underwent revision fusion surgeries, and the remaining one-third (36%, 1059 patients) underwent revisional decompressive surgery without fusion as the first reoperation. We carefully suggest that fusion surgeries were more frequently performed as a method of reoperation because pseudoarthrosis is one of the main causes of early reoperation. To the best of our knowledge, this is the first study to report reoperation rates in all spinal regions. Although numerous studies have reported reoperation rates after spinal surgery, most of them are limited to specific spinal regions, and their results are not useful for comparing the reoperation rates of spinal surgeries performed in different regions. In our study, the third-year reoperation rates were 5.2% (95% CI, [5.0–5.4]) after lumbar spinal fusion, 3.2% (95% CI, [2.2–4.2]) after thoracic spinal fusion, and 2.7% (95% CI, [2.5–3.0]) after cervical spinal fusion (Figure 3). Inferring the causes of the different reoperation rates, especially between cervical and lumbar spinal surgery, is beyond the scope of our study. However, we carefully suggest that the reoperation rate of the lumbar spine is higher than that of the cervical spine because posterior surgical approach was more frequently performed in lumbar spine surgery. As the follow-up time after the index surgery increased, the gaps in reoperation rates among the different spinal regions widened, especially between the lumbar region and other regions (Figure 3).

Reoperation rates were significantly different according to the type of surgical approach and the spinal region. Before comparing reoperation rates by surgical approaches, we excluded the two least common approaches, including the anterior thoracic approach (34 cases in total, 0.05%) and the combined thoracic approach (27 cases in total, 0.04%), since these two approaches are not usual treatment methods for degenerative spinal disease. Further, the two approaches were not sufficient to determine their actual reoperation rates (Table 4). The third-year reoperation rates were as follows: 5.3% (95% CI, [4.6–5.9]) in the combined lumbar approach, 5.2% (95% CI, [5.0–5.4]) in the posterior lumbar approach, 5.0% (95% CI, [3.7–6.3]) in the anterior lumbar approach, 3.0% (95% CI, [2.0–4.0]) in the posterior thoracic approach, 2.8% (95% CI, [1.9–3.8]) in the posterior cervical approach, 2.6% (95% CI, [2.5–3.0]) in the anterior cervical approach, and 1.6% (95% CI, [0.3–2.9]) in the combined cervical approach. Based on these reoperation rates, an approach that shows lower reoperation rates can be considered when diverse surgical approaches can be applied with similar expected clinical outcomes.

In addition, based on a sufficient number of patients using a nationwide database, we presented annual reoperation rates according to diverse patient characteristics (Table 2) and medical comorbidities (Table 3). Although 1st-year reoperation rates were highest (2.2%) in the oldest age group (age > 80 years), 3rd-year reoperation rates were highest (5.3%) in patients aged between 70 and 79 years. Reoperation rates were also higher in patients of male sex and those who live in a rural residence, and in those who received transfusions or had a longer duration of steroid use. Interestingly, although reoperation rates were higher in patients with severe medical comorbidities according to the CCI score, the pattern significantly varied according to the type of comorbidity. Reoperation rates were consistently higher in patients with renal disease and Parkinson’s disease throughout the study period. In contrast, 1st-year reoperation rates were especially high in patients with congestive heart failure (2.9%) and complicated diabetes (2.8%), and 3rd-year reoperation rates were highest in patients with moderate to severe liver disease (11.5%) despite the lower 1st-year reoperation rate (1.6%). It is well known that a prediction by a simple ‘base rate’ of the entire population can be comparable to that obtained from a complex multivariable analysis, in terms of prediction accuracy [18]. Our reoperation rates are based on a nationwide database that includes the entire population, and annual reoperation rates according to a specific factor can be used as a base reoperation rate for the entire population with a specific factor. In other words, the data in Table 2, Table 3 and Table 4 can be used as a reasonable source of base rates, which are important starting points for modern Bayesian statistics.

Based on these differences, we created a prediction model (Model 2 in Table 5) for early reoperation within 3 years after instrumented spinal fusion surgery for degenerative spinal disease. Our prediction model, which is based on the HIRA database, has several advantages. Using data from a large number of patients, we could include core clinical information, such as precise medical comorbidities and diverse surgical approaches, for all spinal regions in our model. In fact, reoperation risks associated with these variables cannot be easily evaluated even in multicenter studies. However, we calculated their adjusted ORs for reoperation (model 2) with greater precision (narrow 95% CI, Table 5), despite including all these variables. Second, we performed a two-step validation procedure for the prediction model. Initially, through an internal validation procedure using the bootstrap method, all estimates of the prediction model were validated with a relative bias based on 1000 bootstrapped samples. Bootstrapping is a sampling method that uses random sampling of cases with replacement. The CIs calculated from bootstrap sampling (1000 times in our study) are asymptotically more accurate than those from the conventional method, and this can reduce the possible skewness of independent variables, including unknown confounders. In addition, we performed a sensitivity analysis based on different time points for reoperation (1st-year reoperation), and potential effect modification by unknown confounders was assessed by comparing the adjusted ORs of the consistent predictors that remained significant in the two prediction models. Most of our predictors for 3rd-year reoperation also remained significant in the prediction model for 1st-year reoperation, and the Bland–Altman test did not show significant effect modification of our prediction model.

Nevertheless, our results should be interpreted carefully considering the following limitations. First, the HIRA database is a claims database that was not originally designed for clinical research. Possible discrepancies between the diagnostic codes in the database and the actual diseases may be potential sources of bias. By contrast, the HIRA system is based on the compulsory national health insurance system, and government officials thoroughly review all claims data using various regulatory protocols. Among them, the control policy for patients who undergo high-revenue spinal surgeries has been the object of priority. Information about drug and device use and precise surgical approaches in patients who undergo instrumented spinal surgeries is thoroughly reviewed by government officials and is thus very accurate. Second, we could not identify the exact causes of reoperation in our cohort. Therefore, reoperation could be performed in the same area due to early failure including pseudoarthrosis or in the adjacent segment due to late causes including adjacent segment degeneration. Third, important information possibly related to the risk for reoperation, such as individual radiologic information, including global or regional sagittal alignment, or precise surgical profiles, including differences in surgical protocols and techniques among hospitals or surgeons, could not be included in the study. Although the two-step validation procedures confirmed the consistency of our results, the inclusion of these variables would be helpful in increasing the prediction power. Finally, we could not include patients with degenerative spinal deformities due to the limited data capacity for analysis. Further studies with improved data capacity and computing power are required to include all types of spinal diseases.

In conclusion, reoperation is a major concern in patients undergoing instrumented spinal fusion surgery for degenerative spinal disease. Our study, based on data from 65,355 population cohorts, clearly presented the annual reoperation rates within 3 years after the index surgery according to surgical regions and approaches, clinical characteristics, and medical comorbidities. Our results, including the base rates and their aggregated prediction model, provide clinicians with an acceptable tool for the comprehensive risk assessment of early reoperation in patients who will undergo instrumented spinal fusion surgery for degenerative spinal disease.

## Figures and Tables

**Figure 1 jcm-11-03338-f001:**
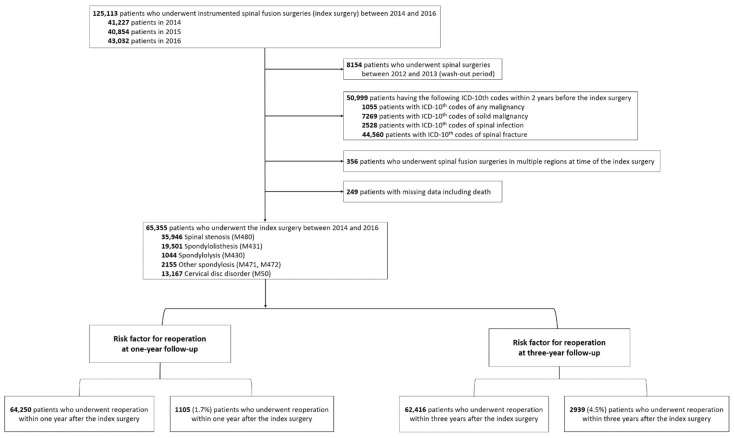
Selection of the study population.

**Figure 2 jcm-11-03338-f002:**
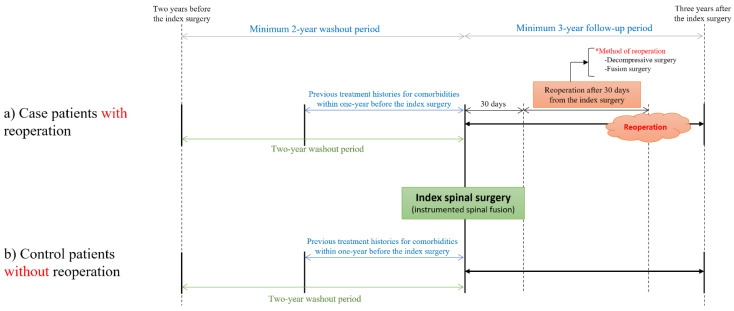
Definitions of cases and controls. * Reoperation was defined as subsequent surgery, which only included decompressive or fusion surgery.

**Figure 3 jcm-11-03338-f003:**
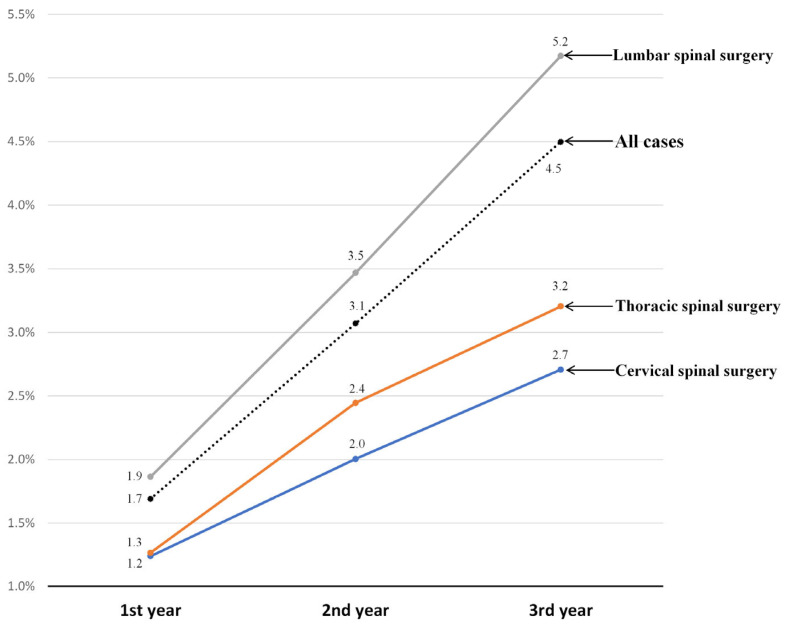
Annual reoperation rates according to the spinal regions.

**Figure 4 jcm-11-03338-f004:**
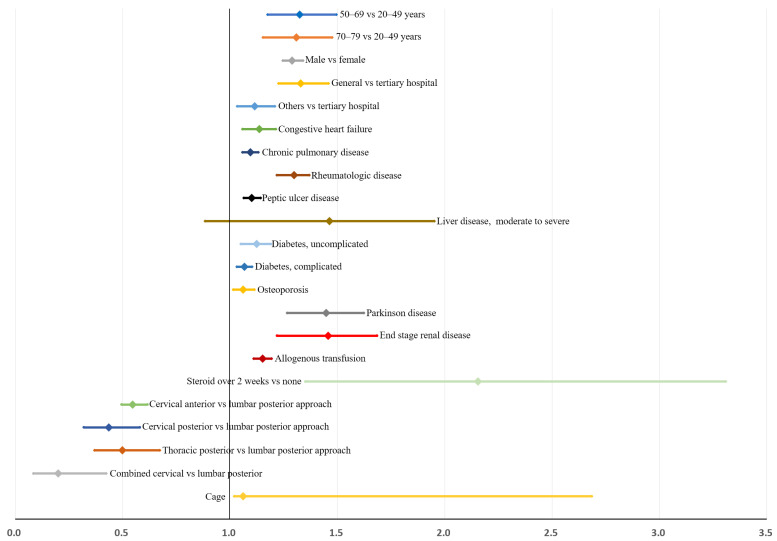
Risk factors for reoperation within 3 years after instrumented spinal fusion surgery for degenerative spinal disease: bootstrap-adjusted odds ratios and their 95% confidence interval (Table 5).

**Figure 5 jcm-11-03338-f005:**
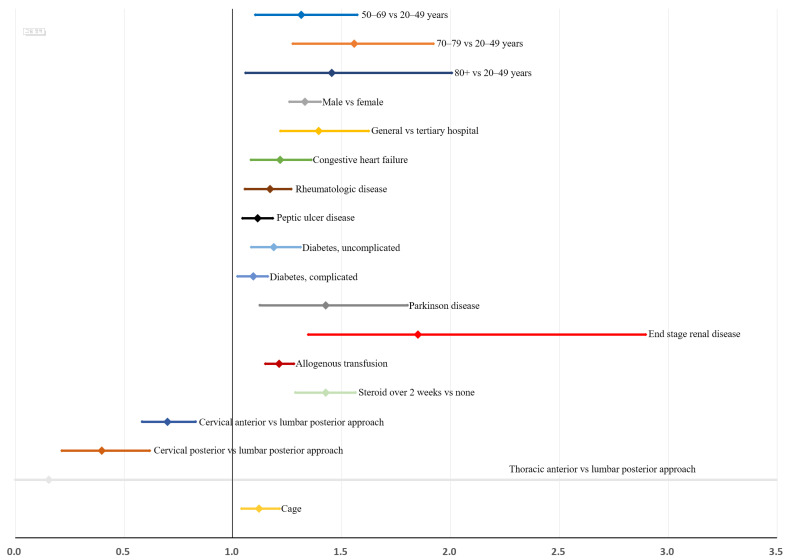
Risk factors for reoperation within 1 year after instrumented spinal fusion surgery for degenerative spinal disease: bootstrap-adjusted odds ratios and their 95% confidence interval (Appendix A).

**Table 1 jcm-11-03338-t001:** Annual reoperation rates according to the spinal regions.

Index Regions	All Cases (%)	Reoperation Cases at 1-Year Follow-Up (*n*)	Reoperation Cases at 2-Year Follow-Up (*n*)	Reoperation Cases at 3-Year Follow-Up (*n*)	Reoperation Rates at the 1st Year	Reoperation Rates at the 2nd Year	Reoperation Rates at the 3rd Year
Cervical spine	16,962 (26)	210	340	459	1.2 [1.1–1.4]	2.0 [1.8–2.2]	2.7 [2.5–3.0]
Thoracic spine	1186 (2)	15	29	38	1.3 [0.6–1.9]	2.4 [1.6–3.3]	3.2 [2.2–4.2]
Lumbar spine	47,207 (72)	880	1637	2442	1.9 [1.7–2.0]	3.5 [3.3–3.6]	5.2 [5.0–5.4]
All regions	65,355	1105	2006	2939	1.7 [1.6–1.8]	3.1 [2.9–3.2]	4.5 [4.3–4.7]

**Table 2 jcm-11-03338-t002:** Annual reoperation rates according to patients’ characteristics.

Variables	Categories	All	Patients with Reoperation within 1 Year after the Index Surgery	Patients with Reoperation within 2 Years after the Index Surgery	Patients with Reoperation within 3 Years after the Index Surgery	Odds Ratios for Reoperation within 3 Years after the Index Surgery	
Number of patients		65,355	1105 (1.7)	2006 (3.1)	2939 (4.5)		
Age	Mean ± SD	61.4 ± 11.5	64.0 ± 11.0	63.8 ± 10.5	63.5 ± 10.4	1.02 (1.01–1.02)	<0.001
	20–49	10,011	109 (1.1)	187 (1.9)	278 (2.8)	Reference	
	50–69	37,446	612 (1.6)	1157 (3.1)	1733 (4.6)	1.70 (1.49–1.93)	<0.001
	70–79	16,058	344 (2.1)	602 (3.7)	849 (5.3)	1.95 (1.70–2.24)	<0.001
	80+	1840	40 (2.2)	60 (3.3)	79 (4.3)	1.57 (1.22–2.03)	0.001
Sex	Male	29,606	608 (2.1)	1084 (3.7)	1511 (5.1)	1.29 (1.20–1.39)	<0.001
	Female	35,749	497 (1.4)	922 (2.6)	1428 (4.0)	Reference	
Region	Urban	55,112	917 (1.7)	1634 (3.0)	2412 (4.4)	Reference	
	Rural	10,243	188 (1.8)	372 (3.6)	527 (5.1)	1.19 (1.08–1.31)	0.001
Hospital	Tertiary	17,767	258 (1.5)	463 (2.6)	680 (3.8)	Reference	
	General	17,166	381 (2.2)	666 (3.9)	942 (5.5)	1.50 (1.32–1.61)	<0.001
	Others	30,422	466 (1.5)	877 (2.9)	1317 (4.3)	1.14 (1.03–1.25)	0.008
Transfusion	Autologous transfusion	313	7 (2.2)	10 (3.2)	18 (5.8)	1.30 (0.81–2.09)	0.285
	Allogenous transfusion	26,142	570 (2.2)	1015 (3.9)	1487 (5.7)	1.57 (1.46–1.69)	<0.001
Systemic steroid	None	46,894	714 (1.5)	1345 (2.9)	1978 (4.2)	Reference	
	within 2 weeks	17,977	386 (2.1)	648 (3.6)	944 (5.3)	1.26 (1.16–1.36)	<0.001
	Over 2 weeks	214	5 (2.3)	13 (6.1)	17 (7.9)	1.96 (1.19–3.22)	<0.001

The numbers in parentheses mean annual reoperation rates.

**Table 3 jcm-11-03338-t003:** Annual reoperation rates according to comorbidities.

Variables	Categories	All	Patients with Reoperation within 1 Year after the Index Surgery	Patients with Reoperation within 2 Years after the Index Surgery	Patients with Reoperation within 3 Years after the Index Surgery	Odds Ratios for Reoperation within 3 Years after the Index Surgery	
Number of patients		65,355	1105 (1.7)	2006 (3.1)	2939 (4.5)		
Charlson comorbidity index score	Mean ± SD	1.63 ± 1.82	2.04 ± 2.02	2.03 ± 2.02	2.05 ± 2.00	1.13 (1.01–1.15)	<0.001
	0–2	45,727	690 (1.5)	1258 (2.8)	1817 (4.0)	Reference	
	3–5	17,292	349 (2.0)	629 (3.6)	949 (5.5)	1.40 (1.30–1.52)	<0.001
	≥6	2336	66 (2.8)	119 (5.1)	173 (7.4)	1.93 (1.64–2.27)	<0.001
Comorbidities	Myocardial infarction	577	15 (2.6)	24 (4.2)	34 (5.9)	1.33 (0.94–1.89)	0.105
	Congestive heart failure	2134	62 (2.9)	100 (4.7)	142 (6.7)	1.54 (1.29–1.93)	<0.001
	Peripheral vascular disease	6712	131 (2.0)	240 (3.6)	351 (5.2)	1.20 (1.07–1.34)	0.002
	Cerebrovascular disease	6083	122 (2.0)	223 (3.7)	333 (5.5)	1.26 (1.12–1.42)	<0.001
	Dementia	554	7 (1.3)	14 (2.5)	25 (4.5)	1.00 (0.67–1.50)	0.986
	Chronic pulmonary disease	15,003	284 (1.9)	531 (3.5)	820 (5.5)	1.32 (1.21–1.43)	<0.001
	Rheumatologic disease	3235	74 (2.3)	162 (5.0)	243 (7.5)	1.79 (1.56–2.05)	<0.001
	Peptic ulcer disease	11,878	249 (2.1)	447 (3.8)	660 (5.6)	1.32 (1.21–1.44)	<0.001
	Liver disease						
	Mild	4151	81 (2.0)	156 (3.8)	220 (5.3)	1.20 (1.05–1.39)	0.010
	Moderate to severe	61	1 (1.6)	3 (4.9)	7 (11.5)	2.76 (1.25–6.06)	0.012
	Diabetes						
	Uncomplicated	14,001	309 (2.2)	553 (3.9)	774 (5.5)	1.33 (1.23–1.45)	<0.001
	Complicated	4375	124 (2.8)	207 (4.7)	285 (6.5)	1.53 (1.35–1.74)	<0.001
	Hemiplegia or paraplegia	654	11 (1.7)	23 (3.5)	27 (4.1)	0.91 (0.62–1.35)	0.648
	Renal disease	1216	49 (4.0)	63 (5.2)	84 (6.9)	1.59 (1.27–1.99)	<0.001
	Osteoporosis	10,026	166 (1.7)	324 (3.2)	502 (5.0)	1.14 (1.04–1.26)	0.007
	Parkinson disease	529	20 (3.8)	40 (7.6)	51 (9.6)	2.29 (1.71–3.06)	<0.001
	End stage renal disease	272	18 (6.6)	23 (8.5)	30 (11.0)	2.65 (1.81–3.88)	<0.001

The numbers in parentheses mean annual reoperation rates.

**Table 4 jcm-11-03338-t004:** Annual reoperation rates according to the surgical procedures.

Variables	Cases (%)	Interval *	Reoperation Rates at 1st Year (%)			Reoperation Rates at 2nd Year (%)			Reoperation Rates at 3rd Year (%)			Annual Reoperation Rates and 95% Confidence Intervals		
			Overall	Revisional Fusion	Revisional Decompression	Overall	Revisional Fusion	Revisional Decompression	Overall	Revisional Fusion	Revisional Decompression	1st Year	2nd Year	3rd Year
All	65,355	520 (232, 800)	1105 (1.7)	752 (1.1)	353 (0.5)	2006 (3.1)	1332 (2.0)	674 (1.0)	2939 (4.5)	1882 (2.9)	1059 (1.6)	1.7 [1.6–1.8]	3.1 [2.9–3.2]	4.5 [4.3–4.7]
Cervical														
Anterior	15,391 (23.5)	418 (173, 745)	193 (1.2)	126 (0.8)	67 (0.4)	310 (1.9)	205 (1.3)	105 (0.7)	419 (2.6)	273 (1.7)	146 (0.9)	1.2 [1.1–1.4]	1.9 [1.8–2.2]	2.6 [2.5–3.0]
Posterior	1202 (1.8)	476 (126, 742)	13 (1.1)	11 (0.9)	2 (0.2)	25 (2.1)	22 (1.8)	3 (0.2)	34 (2.8)	29 (2.4)	5 (0.4)	1.1 [0.5–1.7]	2.1 [1.3–2.9]	2.8 [1.9–3.8]
Thoracic														
Anterior	34 (0.05)	63 (54, 79)	3 (8.8)	1 (2.9)	2 (5.9)	3 (8.8)	1 (2.9)	2 (5.9)	3 (8.8)	1 (2.9)	2 (5.9)	-	-	8.8 [0–18.4]
Posterior	1125 (1.7)	462 (307, 718)	12 (1.1)	11(1.0)	1 (0.1)	26 (2.3)	22 (2.0)	4 (0.4)	34 (3.0)	29 (2.6)	5 (0.4)	1.1 [0.5–1.7]	2.3 [1.4–3.2]	3.0 [2.0–4.0]
Lumbar														
Anterior	1058 (1.6)	527 (94, 777)	24 (2.3)	17 (1.6)	7 (0.7)	37 (3.5)	27 (2.6)	10 (0.9)	53 (5.0)	34 (3.2)	19 (1.8)	2.3 [1.4–3.2]	3.5 [2.4–4.6]	5.0 [3.7–6.3]
Posterior	41,259 (63.1)	540 (245, 818)	773 (1.9)	549 (1.3)	224 (0.5)	1426 (3.5)	989 (2.4)	437 (1.1)	2132 (5.2)	1418 (3.4)	714 (1.7)	1.9 [1.7–2.0]	3.5 [3.3–3.6]	5.2 [5.0–5.4]
Combined approach #														
Cervical	369 (0.6)	160 (57, 671)	4 (1.1)	3 (0.8)	1 (0.3)	5 (1.4)	4 (1.1)	1 (0.3)	6 (1.6)	5 (1.4)	1 (0.3)	1.1 [0–2.1]	1.4 [0.2–2.5]	1.6 [0.3–2.9]
Thoracic	27 (0.04)	777 (single case)	0	0	0	0	0	0	1 (3.7)	1 (3.7)	0 (0)	-	-	3.7 [0–10.8]
Lumbar	4890 (7.5)	557 (289, 800)	83 (1.7)	34 (0.7)	49 (1.0)	174 (3.6)	62 (1.3)	112 (2.3)	257 (5.3)	92 (1.9)	167 (3.4)	1.7 [1.3–2.1]	3.6 [3.0–4.1]	5.3 [4.6–5.9]
Cage	30,977 (47.4)	560 (258, 826)	530 (1.7)	364 (1.2)	166 (0.5)	1011 (3.3)	686 (2.2)	325 (1.0)	1531 (4.9)	996 (3.2)	535(1.7)	1.7 [1.6–1.9]	3.3 [3.1–3.5]	4.9 [4.7–5.2]

* indicates the interval (days) between the index surgery and reoperation and is presented with interquartile ranges. # means combined anterior and posterior approach.

**Table 5 jcm-11-03338-t005:** Risk factors for reoperation within 3 years after instrumented spinal fusion surgery for degenerative spinal disease: multivariable analysis.

Variables	Categories	Model 1		Model 2 (Backward)		Model 3 (Bootstrap Adjusted)	Bias (%)
		Adjusted Odds Ratio (95% Confidence Interval)	*p*-Value	Adjusted Odds Ratio (95% Confidence Interval)	*p*-Value	Adjusted Odds Ratio (95% Confidence Interval)	
Age	50–69 vs. 20–49 years	1.30 (1.14–1.50)	<0.001	1.32 (1.15–1.51)	<0.001	1.32 (1.18–1.50)	−0.3%
	70–79 vs. 20–49 years	1.28 (1.10–1.49)	0.002	1.30 (1.12–1.51)	0.001	1.31 (1.15–1.48)	−2.3%
	80+ vs. 20–49 years	1.01 (0.77–1.31)	0.963				
Sex	Male vs. female	1.66 (1.53–1.80)	<0.001	1.66 (1.53–1.80)	<0.001	1.29 (1.25–1.34)	22.3%
Region	Rural vs. urban	1.10 (1.00–1.21)	0.065				
Hospital	General vs. tertiary	1.31 (1.18–1.46)	<0.001	1.33 (1.20–1.48)	<0.001	1.33 (1.23–1.46)	0.1%
	Others vs. tertiary	1.12 (1.01–1.23)	0.028	1.12 (1.02–1.23)	0.023	1.12 (1.03–1.46)	3.4%
Charlson comorbidity index score	3–5 vs. 0–2	1.04 (0.85–1.27)	0.700				
	≥6 vs. 0–2	0.92 (0.64–1.33)	0.646				
Comorbidities	Congestive heart failure	1.30 (1.08–1.56)	0.005	1.29 (1.08–1.54)	0.005	1.14 (1.06–1.21)	49.8%
	Peripheral vascular disease	1.04 (0.92–1.17)	0.538				
	Cerebrovascular disease	1.10 (0.97–1.24)	0.154				
	Chronic pulmonary disease	1.63 (0.88–3.02)	0.123	1.20 (1.10–1.30)	<0.001	1.10 (1.06–1.13)	50.2%
	Rheumatologic disease	1.70 (1.48–1.97)	<0.001	1.69 (1.47–1.94)	<0.001	1.30 (1.22–1.37)	50.1%
	Peptic ulcer disease	1.21 (1.10–1.33)	<0.001	1.21 (1.10–1.33)	<0.001	1.10 (1.06–1.14)	49.2%
	Liver disease, mild	1.10 (0.95–1.28)	0.199				
	Liver disease, moderate to severe	2.40 (1.06–5.42)	0.035	2.41 (1.09–5.34)	0.031	1.46 (0.88–1.95)	56.8%
	Diabetes, uncomplicated	1.14 (1.04–1.26)	0.006	1.14 (1.05–1.26)	0.004	1.12 (1.05–1.19)	11.1%
	Diabetes, complicated	1.27 (1.06–1.53)	0.010	1.26 (1.10–1.44)	<0.001	1.07 (1.03–1.10)	71.6%
	Renal disease	1.00 (0.74–1.35)	0.983				
	Osteoporosis	1.12 (1.01–1.25)	0.038	1.12 (1.01–1.25)	0.034	1.06 (1.01–1.11)	5.2%
	Parkinson disease	2.07 (1.54–2.79)	<0.001	2.10 (1.56–2.82)	<0.001	1.45 (1.27–1.62)	31.1%
	End stage renal disease	2.19 (1.36–3.52)	0.001	2.15 (1.45–3.17)	<0.001	1.46 (1.22–1.69)	32.2%
Allogenous transfusion		1.32 (1.22–1.44)	<0.001	1.33 (1.23–1.45)	<0.001	1.15 (1.11–1.19)	13.3%
Systemic steroid	within 2 weeks vs. none	1.30 (1.12–1.40)	0.278				
	Over 2 weeks vs. none	2.25 (1.33–3.81)	0.011	2.25 (1.33–3.80)	0.011	2.15 (1.35–3.31)	4.2%
Surgical approach	Cervical anterior vs. lumbar posterior	0.55 (0.48–0.63)	<0.001	0.55 (0.48–0.63)	<0.001	0.55 (0.50–0.62)	0.4%
	Cervical posterior vs. lumbar posterior	0.45 (0.32–0.64)	<0.001	0.45 (0.31–0.64)	<0.001	0.44 (0.32–0.58)	3.3%
	Thoracic posterior vs. lumbar posterior	0.51 (0.36–0.72)	<0.001	0.51 (0.36–0.72)	<0.001	0.50 (0.37–0.67)	2.3%
	Combined cervical vs. lumbar posterior	0.22 (0.10–0.50)	<0.001	0.22 (0.09–0.50)	<0.001	0.20 (0.08–0.42)	9.0%
Cage		1.12 (1.02–1.23)	0.021	1.13 (1.02–1.24)	0.014	1.06 (1.02–2.69)	6.1%

All significant independent variables (*p* < 0.05) from the univariable analysis were initially included and subsequently selected by backward stepwise selection in model 2. Relative bias was estimated as the difference between the mean bootstrapped regression coefficient estimates (Model 3) and the mean parameter estimates of Model 2 divided by the mean parameter estimates of Model 2.

**Table 6 jcm-11-03338-t006:** Comparisons of the adjusted odds ratios of the consistent variables in the two prediction models (1st-year vs. 3rd-year reoperation).

		Odds Ratios for Model 2 of 3rd-Year Reoperation		Odds Ratios for Model 2 of 1st-Year Reoperation		Bland–Altman Test for Interaction
Age	50–69 vs. 20–49 years	1.32 (1.15–1.51)	<0.001	1.31 (1.06–1.63)	0.014	0.953
	70–79 vs. 20–49 years	1.30 (1.12–1.51)	0.001	1.54 (1.21–1.95)	<0.001	0.238
Sex	Male vs. female	1.66 (1.53–1.80)	<0.001	1.79 (1.58–2.03)	<0.001	0.322
Hospital	General vs. tertiary	1.33 (1.20–1.48)	<0.001	1.39 (1.18–1.63)	<0.001	0.653
Comorbidities	Congestive heart failure	1.29 (1.08–1.54)	0.005	1.46 (1.12–1.90)	0.005	0.463
	Rheumatologic disease	1.69 (1.47–1.94)	<0.001	1.35 (1.06–1.72)	0.016	0.115
	Peptic ulcer disease	1.21 (1.10–1.33)	<0.001	1.24 (1.08–1.44)	0.003	0.781
	Diabetes, uncomplicated	1.14 (1.05–1.26)	0.004	1.20 (1.04–1.38)	0.013	0.550
	Diabetes, complicated	1.26 (1.10–1.44)	<0.001	1.39 (1.13–1.70)	0.002	0.431
	Parkinson disease	2.10 (1.56–2.82)	<0.001	2.03 (1.29–2.20)	0.002	0.868
	End stage renal disease	2.15 (1.45–3.17)	<0.001	3.03 (1.85–4.98)	<0.001	0.287
Allogenous transfusion		1.33 (1.23–1.45)	<0.001	1.47 (1.28–1.69)	<0.001	0.224
Systemic steroid	Over 2 weeks vs. none	2.25 (1.33–3.80)	0.011	1.43 (1.26–1.62)	<0.001	0.100
Surgical approach	Cervical anterior vs. lumbar posterior	0.55 (0.48–0.63)	<0.001	0.69 (0.56–0.86)	0.001	0.080
	Cervical posterior vs. lumbar posterior	0.45 (0.31–0.64)	<0.001	0.42 (0.24–0.74)	0.003	0.840
Cage		1.13 (1.02–1.24)	0.014	1.27 (1.09–1.47)	0.002	0.596

## Data Availability

The datasets generated for the current study are not publicly available due to Data Protection Laws and Regulations in Korea, but the analyzing results are available from the corresponding authors on reasonable request.

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
