# Peer review of "Early Reoperation Rates and Its Risk Factors after Instrumented Spinal Fusion Surgery for Degenerative Spinal Disease: A Nationwide Cohort Study of 65,355 Patients"

_jcm, 2022, doi:10.3390/jcm11123338_

Round 1
Reviewer 1 Report
The authors aimed to present reasonable base rates for reoperation according to all possible risk factors and build a validated prediction model for early reoperation. This paper is well written and presentation is very good
Author Response
Thank you very much for your kind comments regarding our study.
Reviewer 2 Report
Kim et al. propose a multivariable analysis for detection of reoperation rate in spinal fusion surgery.
Introduction
The rationale to perform the analyses is underdeveloped. Please try to give the reader a concise hypothesis in the end of the introduction.
Materials and Methods
Why 30 days after the index surgery? This does not cover long-term complications, and especially some infections might occur later on.
Results
Table 1-4 seem of value for the reader, not not the main focus. Subsequently, the results section should be substantially shortened.
The final paragraph is of crucial importance for the whole manuscript. Please explain your findings in more details.
Discussion
A clinical consequence of your findings would be highly appreciated.
Author Response
Kim et al. propose a multivariable analysis for the detection of reoperation rate in spinal fusion surgery.
à Thank you very much for your invaluable comments regarding our study
- Introduction
The rationale to perform the analyses is underdeveloped. Please try to give the reader a concise hypothesis in the end of the introduction.
à Thank you for your suggestion. The rationale to perform our analysis was originally presented in the introduction section. To present the rationale more clearly, we revised the introduction as follows.
The risk of reoperation is influenced by numerous factors, including the patients’ symptoms, radiological findings of the disease, and diverse factors, including lifestyle, health insurance, surgeon preference, and contemporary treatment trends. Some of these factors cannot be easily quantified, and they inevitably act as confounders in studies investigating reoperation risks. It is impossible to avoid the effects of these confounders; hence, researchers generally adopt one of the following two strategies: the first strategy is to report the reoperation rates of study cohorts with a large sample size, and the second is to build a prediction model for reoperation using all possible information. To successfully implement these strategies, studies reporting reoperation rates should be based on the general population, and studies on building a prediction model for reoperation should be statistically validated considering inevitable biases from confounders. However, these two conditions have not been achieved in previous studies. The reported reoperation rates based on limited samples were inevitably confined to a specific spinal region, mainly the cervical or lumbar regions, without a comprehensive analysis including all spinal regions.[3,4,6-9] Therefore, reoperation rates and risk factors vary considerably according to the characteristics of their study patients and design.[7,9-11] Discrepancies in the estimates of unvalidated prediction models exponentially increase when calculating the risk factors for long-term reoperation.[7,12-15]
To overcome the limitations of previous studies, we conducted a population-based cohort study using nationwide claims data. Our study has two distinct research purposes. First, using a database including the entire population and their surgical records across all spinal regions, we aimed to present reasonable ‘base rates’ for reoperation according to all possible risk factors obtained from the database. The risk for morbidity from reoperation sharply increases when primary surgery is more invasive, and reoperation rates in patients who undergo more invasive primary surgery are clinically important. Therefore, we selected patients who had undergone instrumented fusion surgery for degenerative spinal disease. Second, based on these population cohorts, we tried to build a validated prediction model for early reoperation, since a prediction model for long-term reoperation is greatly influenced by numerous unknown confounders and is theoretically incapable of making accurate predictions.
- Materials and Methods
Why 30 days after the index surgery? This does not cover long-term complications, and especially some infections might occur later on.
à The 30-day period was set up to identify the “staged” surgery including combined anterior and posterior approaches. In our country, patients who are scheduled to undergo staged surgery often undergo their second-stage operation 1-3 weeks after the first operation. Actually, it was difficult to distinguish such staged operation from reoperation in all patients; hence, we set up such period to define reoperation as operation performed 30 days after the index surgery.
In addition, reoperation was defined as a subsequent operation only including decompressive or fusion surgery in our study (Figure 2). It is very rare for such a subsequent operation to be performed within 30 days after the index surgery.
To present the definition of reoperation more clearly, we revised Figure 2 as follows.
Figure 2. Definitions of cases and controls (attachment)
- Results
1) Table 1-4 seem of value for the reader, not the main focus. Subsequently, the results section should be substantially shortened.
à Thank you for your comment. As we stated in the introduction section, our study has two distinct research purposes.
First, using a database including the entire population and their surgical records across all spinal regions, we aimed to present reasonable ‘base rates’ for reoperation according to all possible risk factors obtained from the database. The risk for morbidity from reoperation sharply increases when primary surgery is more invasive, and reoperation rates in patients who undergo more invasive primary surgery are clinically important. Therefore, we selected patients who had undergone instrumented fusion surgery for degenerative spinal disease. Second, based on these population cohorts, we tried to build a validated prediction model for early reoperation, since a prediction model for long-term reoperation is greatly influenced by numerous unknown confounders and is theoretically incapable of making accurate predictions.
We presented the “base rates” for reoperation according to all possible risk factors in Table 1, 2, 3 and 4. Presenting such base rates was one of the main purposes of our study.
2) The final paragraph is of crucial importance for the whole manuscript. Please explain your findings in more detail
à Thank you very much for your comment. The final paragraph of the result section is about the second step of the validation process, as follows.
3.5.2. Sensitivity analysis and Bland-Altman test for interaction
Sensitivity analysis was performed to assess our prediction model when early reoperation was defined as operation performed within 1-year after the index surgery (Supplementary Table 4). Other types of hospitals, combined cervical approach, and medical comorbidities, including moderate to severe liver disease, chronic pulmonary disease, and osteoporosis, which were significant predictors for the third-year reoperation, were not significant predictors for the first-year reoperation. The bootstrap-adjusted ORs and 95% CIs for the prediction model for the first-year reoperation are displayed in Figure 6. Potential effect modification by unknown confounders was assessed by comparing the adjusted ORs and 95% CIs of the consistent predictors in the two prediction models (models for first-year and third-year reoperation), and are presented in Table 6. No significant effect modification was observed for any variable.
As you commented, this second step of the validation process is important for our study. However, according to our two research purposes (presenting the base rates and prediction model for reoperation), we would like to emphasize the paragraphs with the following subheadings.
3.1. Annual reoperation rates according to the spinal regions (base rates)
3.2. Annual reoperation rates according to patients' characteristics (base rates)
3.3. Annual reoperation rates according to comorbidities (base rates)
3.4. Annual reoperation rates according to the surgical procedures (base rates)
3.5. Risk factors for early reoperation within 3 years after instrumented spinal fusion surgeries: multivariable analysis (prediction model for reoperation)
- Discussion
A clinical consequence of your findings would be highly appreciated.
à Thank you very much for your kind and invaluable comments regarding our study. A statement on the clinical implication has been added to the last paragraph of the Discussion section.

Reviewer 3 Report
I understand that national data on reoperation rates for spinal fusion surgery would be epidemiologically useful. I also believe that the information on reoperation rates by site and by approach is valuable to be able to compare within a single study.
However, there are some points of concern from the spine clinician's perspective.
(1) Differences in reoperation rates by site
In your study, lumbar spine surgery resulted in a higher reoperation rate than cervical spine surgery. (The odds ratio is low.) Do you have any discussion on the reasons for this?
2) Regarding the difference in the rates of fusion and decompression as reoperations
For both cervical and lumbar spine surgeries, fusion accounted for the majority of reoperations. Do you have any discussion on the reasons for this?
(iii) Regarding the spinal level at which reoperation was performed.
The clinical intent of reoperation is completely different whether the same area is operated on again because the implant loosening was occurred, etc., or whether an additional procedure was performed for a lesion at another vertebral level that was not a problem initially. Have you evaluated this point?
4) Comorbidities that are risk factors for reoperation
What new results, if any, have you found that have not been shown in prior studies? (Are the results of this study similar to previous results?)
Author Response
Response to the third reviewer’s comments
I understand that national data on reoperation rates for spinal fusion surgery would be epidemiologically useful. I also believe that the information on reoperation rates by site and by approach is valuable to be able to compare within a single study. However, there are some points of concern from the spine clinician's perspective.
à Thank you very much for your kind and invaluable comments regarding our study.
1) Differences in reoperation rates by site
In your study, lumbar spine surgery resulted in a higher reoperation rate than cervical spine surgery. (The odds ratio is low.) Do you have any discussion on the reasons for this?
à Thank you very much for your valuable comment and question. Presenting the causes of the different reoperation rates is beyond the scope of our study. However, it can be reasonably explained by the difference in the method of surgical approach. Generally, incidence of adjacent segment degeneration after spinal fusion surgery is higher in the posterior approach than in the anterior approach. In our cohort, posterior surgical approach was predominantly performed in lumbar fusion surgeries and anterior surgical approach was predominantly performed in cervical fusion surgeries. Therefore, we carefully suggest that one of the major causes of the different reoperation rates between cervical and lumbar fusion surgeries is the difference in adjacent segment degeneration by surgical approach. According to your suggestion, we revised the discussion as follows.
…… after cervical spinal fusion (Figure 3). Inferring the causes of the different reoperation rates, especially between cervical and lumbar spinal surgery, is beyond the scope of our study. However, we carefully suggest that the reoperation rate of the lumbar spine is higher than that of the cervical spine because posterior surgical approach was more frequently performed in lumbar spine surgery. As the follow-up time after the index surgery increased, the gaps in reoperation rates among the different spinal regions widened, especially between the lumbar region and other regions (Figure 3).
2) Regarding the difference in the rates of fusion and decompression as reoperations
For both cervical and lumbar spine surgeries, fusion accounted for the majority of reoperations. Do you have any discussion on the reasons for this?
à We appreciate this excellent question. We included only the patients who underwent instrumented spinal fusion surgeries for degenerative spinal disease. Although the exact causes of reoperation could not be identified in our cohort, fusion surgeries are thought to be more frequently performed as a method of reoperation because pseudoarthrosis is one of the main causes of early reoperation. According to your suggestion, we revised the discussion section follows.
Among them, approximately two-thirds (64%, 1,882 patients) underwent revision fusion surgeries, and the remaining one-third (36%, 1,059 patients) underwent revisional decompressive surgery without fusion as the first reoperation. We carefully suggest that fusion surgeries were more frequently performed as a method of reoperation because pseudoarthrosis is one of the main causes of early reoperation.
3) Regarding the spinal level at which reoperation was performed.
The clinical intent of reoperation is completely different whether the same area is operated on again because the implant loosening was occurred, etc., or whether an additional procedure was performed for a lesion at another vertebral level that was not a problem initially. Have you evaluated this point?
à Thank you for this important question regarding our study. As we mentioned, we could not identify the exact causes of reoperation in our cohort. Therefore, reoperation could be performed in the same area due to early failure including pseudoarthrosis or in the adjacent segment due to late causes including adjacent segment degeneration. We have added this statement as part of the limitations of our study as follows.
Second, we could not identify the exact causes of reoperation in our cohort. Therefore, reoperation could be performed in the same area due to early failure including pseudoarthrosis or in the adjacent segment due to late causes including adjacent segment degeneration. Third, important information possibly……
4) Comorbidities that are risk factors for reoperation
What new results, if any, have you found that have not been shown in prior studies? (Are the results of this study similar to previous results?)
à To the best of our knowledge, most studies did not investigate the association between precise comorbidities and reoperation (References were added as follows).
The Long-term Reoperation Rate Following Surgery for Lumbar Stenosis: A Nationwide Sample Cohort Study With a 10-year Follow-up. Spine (Phila Pa 1976). 2020 Sep 15;45(18):1277-1284.
Increased Proportion of Fusion Surgery for Degenerative Lumbar Spondylolisthesis and Changes in Reoperation Rate: A Nationwide Cohort Study With a Minimum 5-Year Follow-up. Spine (Phila Pa 1976). 2019 Mar 1;44(5):346-354.
Reoperation Rates Following Instrumented Lumbar Spine Fusion. Spine (Phila Pa 1976). 2018 Feb 15;43(4):295-301.
Increased Volume of Surgery for Lumbar Spinal Stenosis and Changes in Surgical Methods and Outcomes: A Nationwide Cohort Study with a 5-Year Follow-Up. World Neurosurg. 2018 Nov;119:e313-e322.
Only presence of comorbidities was identified as a risk factor for reoperation in the following article:
Reoperation Rates After Surgery for Degenerative Cervical Spine Disease According to Different Surgical Procedures: National Population-based Cohort Study. Spine (Phila Pa 1976). 2016 Oct 1;41(19):1484-1492.
A specific type of disease (diabetes) or patient’s status (status of dialysis) was suggested as a possible risk factor for reoperation in the following articles:
Risk Factors Associated with Readmission and Reoperation in Patients Undergoing Spine Surgery. World Neurosurg. 2018 Feb;110:e627-e635.
The relationship between diabetes and the reoperation rate after lumbar spinal surgery: a nationwide cohort study. Spine J. 2015 May 1;15(5):866-74.
Dialysis is an independent risk factor for perioperative adverse events, readmission, reoperation, and mortality for patients undergoing elective spine surgery. Spine J. 2018 Nov;18(11):2033-2042.